# Enhancer of zeste homolog 2 (Ezh2) is required in mouse Isl1-expressing progenitors for proper development of cardiac and skeletal hindlimb structures

Hamna Ammar[1,2], Daniel B. Patolsky[1,3], Lijun Chi[1], Amalia N. Caballero[1,3] and Paul Delgado-Olguín[1,3,4,*]

## ABSTRACT

Enhancer of zeste homolog 2 (EZH2), the catalytic subunit of Polycomb Repressive Complex 2, catalyses H3K27 trimethylation and directs lineage development programs, yet its function in early multipotent progenitors remains incompletely understood. ISL1 marks cardiopharyngeal and neural crest progenitors that contribute to second heart field derivatives and to craniofacial and limb morphogenesis. Here, Isl1-cre lineage tracing showed that lineage derivatives populated the cardiopharyngeal region and the posteromedial hindlimb bud during mid-gestation, and contributed extensively to the developing right ventricle, outflow tract, and hindlimb. Loss of Ezh2 in Isl1-expressing progenitors resulted in a high incidence of congenital heart defects, dominated by incomplete atrial and ventricular septation and outflow tract malformations, including double outlet right ventricle and persistent *truncus arteriosus*. These defects were accompanied by ventricular wall thickening at birth. In parallel, *Ezh2* mutants exhibited hindlimb skeletal abnormalities, including pelvic and tarsal malformations with impaired ossification. These abnormalities were associated with cyanosis and perinatal demise. Together, these findings identify a requirement for EZH2 in Isl1-lineage contribution to heart and hindlimb development.

**KEY WORDS: Ezh2, Cardiac progenitors, Second heart field, Heart development, Congenital heart defects, Coordinated cardiac and limb morphogenesis**

## INTRODUCTION

Congenital heart defects (CHDs) are the most common congenital anomalies, identified in approximately eight per 1000 live births worldwide (van der Linde et al., 2011). CHDs often occur with extracardiac malformations, including skeletal abnormalities and can present as multisystem developmental syndromes (Goldmuntz et al., 2011; Sifrim et al., 2016). Genetic studies have linked variants in subunits of the Polycomb Repressive Complex 2 (PRC2) to syndromes such as Weaver, Cohen–Gibson, and Imagawa–Matsumoto, which are characterized by skeletal overgrowth, distinctive craniofacial features, and neurodevelopmental abnormalities. CHDs have also been reported in several cases (Cyrus et al., 2019; Gibson et al., 2012; Sarigul et al., 1999; Tatton-Brown et al., 2017).

Enhancer of zeste homolog 2 (Ezh2), the catalytic component of PRC2, is an evolutionarily conserved histone methyltransferase that tri-methylates lysine 27 on histone H3 (H3K27me3) (Cao et al., 2002). Ezh2 regulates gene expression programs required for differentiation and lineage commitment during embryogenesis (Bracken et al., 2006). Consistent with an essential developmental requirement, Ezh2-null embryos fail to complete gastrulation and die before embryonic day 10.5 (O'Carroll et al., 2001). In humans, *EZH2* variants are associated with Weaver syndrome (Gibson et al., 2012). Ezh2 is required in lineage-restricted progenitor populations that contribute to hindlimb and heart development. Conditional deletion of Ezh2 in paired related homeobox 1 (PRRX1)-expressing mesenchyme produces patterning defects and enhances osteogenesis, consistent with Ezh2 constraining osteogenic commitment while supporting expansion of osteoprogenitor populations (Dudakovic et al., 2015). In the cardiovascular system, deletion of Ezh2 in anterior heart field progenitors, or Nkx2-5-labeled progenitors, leads to myocardial hypertrophy (Delgado-Olguin et al., 2012), defective coronary development (Jiang et al., 2023), and impaired endocardial cushion formation with septal defects (Chen et al., 2012; He et al., 2012). In addition, deletion of Ezh2 in Tie2-expressing endothelial cells and the fetal liver niche disrupts vascular integrity and hematopoietic support, resulting in embryonic lethality (Delgado-Olguin et al., 2014; Neo et al., 2018). This evidences Ezh2 requirement across multiple lineages.

LIM homeodomain transcription factor Isl1 marks multipotent progenitors of the splanchnic mesoderm that contribute to the right ventricle and outflow tract, and musculoskeletal structures in mice (Cai et al., 2003; Moretti et al., 2006). Given the central contribution of Isl1-positive second heart field progenitors to outflow tract morphogenesis and cardiac septation, perturbations within this lineage result in conotruncal and septation defects (Gao et al., 2019; Moretti et al., 2006; Osoegawa et al., 2014). Isl1 is also expressed in hindlimb progenitors within the lateral plate mesoderm and is required for hindlimb initiation and posterior patterning (Yang et al., 2006). However, how Ezh2 requirements within early progenitor populations intersect with multisystem congenital phenotypes remains incompletely defined.

Here, we used cre-lox recombination to inactivate Ezh2 in Isl1-expressing progenitors and their derivatives. Consistent with the activity of Isl1-cre, we found that loss of Ezh2 in the Isl1 lineage causes CHDs and hindlimb skeletal defects, revealing requirement for Ezh2 in Isl1-lineage contribution to cardiac and hindlimb morphogenesis.

[1]Translational Medicine, The Hospital for Sick Children, Toronto, ON M5G0A4, Canada. [2]Institute of Medical Sciences, University of Toronto, Toronto, ON M5S1A8, Canada. [3]Department of Molecular Genetics, University of Toronto, Toronto, ON M5S1A8, Canada. [4]Heart and Stroke Richard Lewar Centre of Excellence in Cardiovascular Research, Toronto, ON M5S3H2, Canada.

*Author for correspondence (paul.delgadoolguin@sickkids.ca)

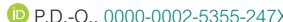 P.D.-O., 0000-0002-5355-247X

## RESULTS

### Derivatives of Isl1-expressing progenitors contribute to the mouse heart, nervous system, and hindlimbs

To validate the activity of Isl1-cre in *Isl1*-expressing progenitors and their derivatives, we crossed *Isl1-cre* transgenic male mice with females carrying the *ROSA26^{mTmG}* reporter, which expresses tdTomato constitutively, and EGFP upon cre-mediated recombination (Muzumdar et al., 2007). Visualization of *Isl1-cre;ROSA26^{mTmG/+}* embryos at day 10.5 (E10.5) and E12.5 under fluorescence microscopy revealed EGFP in derivatives of the neural crest, second heart field, and hind limb buds, consistent with known contributions of Isl1-expressing progenitors. At E10.5, EGFP was present in the neural tube, nerve fibres in branchial arches I to III, and the gonadal region extending distally toward the posterior portion of the hind limb bud (Fig. 1A). Immunofluorescence on embryo sections revealed EGFP positive cells in the outflow tract, differentiating endothelial cells in the endocardial cushions and the right ventricle, and much less prominently toward the left ventricle. EGFP-positive cells were detected discontinuously in the

epicardium and were abundant in the neural epithelium (Fig. 1B). In the hind limb bud, positive cells more prominently occupied the posterior region (Fig. 1B). At E12.5, EGFP fluorescence was detected in the entire right ventricle and outflow tract, and sparsely in the left ventricle. In the hind limb, EGFP signal in the posterior region extended distally toward the developing digits (Fig. 1C).

### Ezh2 inactivation in Isl1-expressing progenitors leads to congenital heart defects and postnatal ventricular wall thickening

Ezh2 function in a population of mouse cardiac progenitors labelled by Nkx2-5-cre that contributes to ventricular cardiomyocytes, and less predominantly to cardiac endothelial derivatives, is required for cardiogenesis (Chen et al., 2012; He et al., 2012). Ezh2 inactivation with Nkx2-5-cre impaired cardiomyocyte proliferation and compaction of the embryonic myocardium, and led to septal defects associated with defective endothelial to mesenchymal transition (Chen et al., 2012; He et al., 2012). This suggests a requirement for Ezh2 beyond

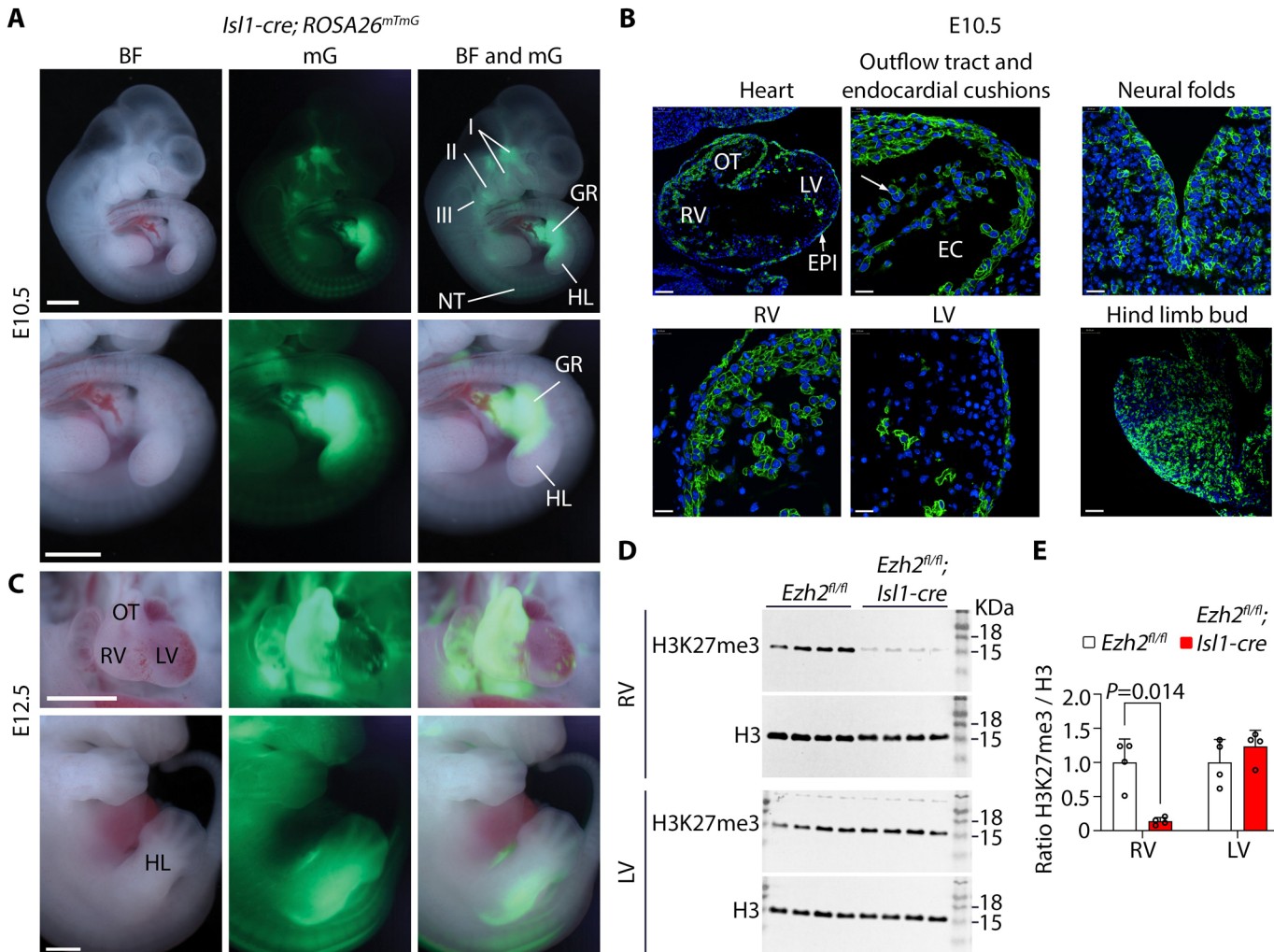

**Fig. 1. Isl1-cre labels cardiac neural crest, second heart field, and hindlimb derivatives.** (A) Representative brightfield (BF) and fluorescence images showing membrane EGFP (mG) in *ROSA26^{mTmG}*;*Isl1*-cre embryos. At E10.5, EGFP localized in the pharyngeal arches I-III, hindlimb (HL) bud, the gonadal region (GR), and neural tube (NT). Scale bars: 1500 μm. (B) Immunofluorescent sections of E10.5 embryos showing EGFP in the outflow tract (OT) and right ventricle (RV), and less toward the left ventricle (LV). EGFP was also detected in epicardial cells (EPI), endothelial cells lining the endocardial cushions (EC), neural folds, and the posterior region of the HL bud. Scale bars: 80 μm (heart) and 20 μm (RV, LV, outflow tract, neural folds, and hindlimb bud). (C) Images of E12.5 embryos showing strong EGFP signal in the right ventricle (RV), OT, the posterior HL region, and in a few cell patches in the LV. Scale bars: 1500 μm. (D) Western blot for H3K27me3 and H3 as loading control on RV and LV of control (*Ezh2^{fl/fl}*) and *Ezh2* mutant (*Ezh2^{fl/fl};Isl1-cre*) embryos at E16.5. (E) Ratio of H3K27me3 / H3. Error bars denote the mean±s.d. of four hearts per genotype, two-tailed unpaired *t*-test with Welch's correction.

cardiomyocytes, including non-myocyte lineages that contribute to outflow tract morphogenesis and endocardial cushion development. To test this, we inactivated Ezh2 in Isl1-expressing progenitors. Mice carrying *loxP* sites targeting the Ezh2 SET domain (O'Carroll et al., 2001) (*Ezh2^{fl/fl}*) were crossed with *Isl1-cre* transgenics (Yang et al., 2006). Ezh2 was specifically inactivated in derivatives of the Isl1-cre lineage. Consistent with activity of the *ROSA26^{mTmG}* reporter (Fig. 1C), Ezh2 and H3K27me3 were decreased in the right, but not the left ventricle of *Ezh2* mutant hearts (*Ezh2^{fl/fl};Isl1-cre*) (Fig. 1D,E;

Fig. S1A,B). In contrast, levels of ubiquitinated lysine 119 of H2A, a PRC1-catalized mark that cooperates with H3K27me3 in gene silencing (Mei et al., 2021), were not affected (Fig. S1C,D).

Hearts of control (*Ezh2^{fl/fl}*) and mutant embryos at E16.5 were analysed by histology. *Ezh2* mutants developed outflow tract defects such as double outlet right ventricle (DORV) and persistent *truncus arteriosus* (PTA) (Fig. 2A,B). In addition, mutant hearts commonly exhibited ventricular septal defects (VSD) and atrial septal defects (ASD) (Fig. 2C, Table 1). These defects were also

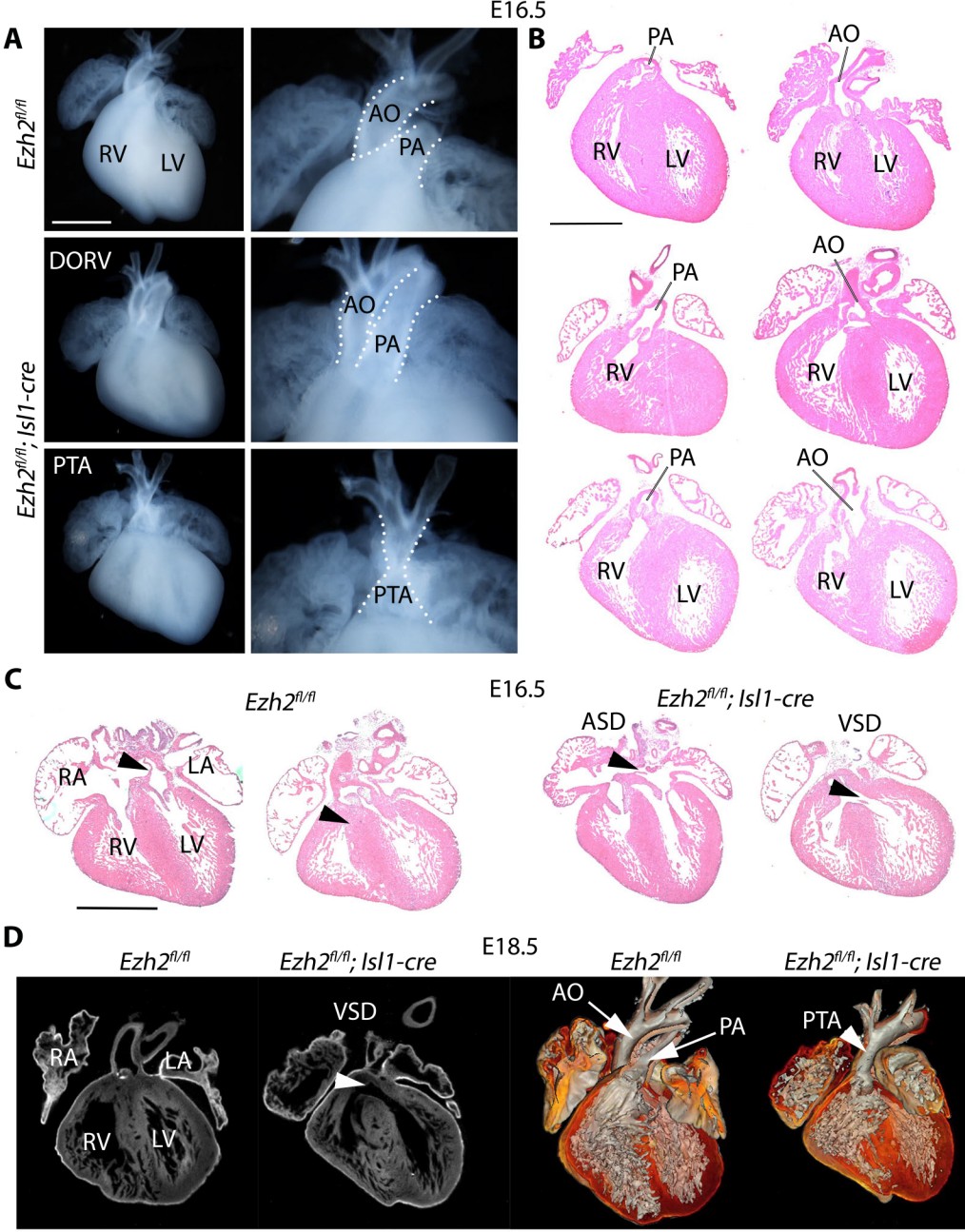

**Fig. 2. *Ezh2* inactivation in Isl1-expressing progenitors causes conotruncal and septation defects.** (A) Gross morphology of representative E16.5 hearts from control (*Ezh2^{fl/fl}*) and mutant (*Ezh2^{fl/fl};Isl1-cre*) embryos. Control hearts show normal outflow tract anatomy, whereas mutant hearts show conotruncal malformations, including double outlet right ventricle (DORV) and persistent truncus arteriosus (PTA). Higher magnification views of the outflow tract region are shown to the right. Scale bar: 500 µm. (B) Haematoxylin and Eosin-stained transverse sections of E16.5 control and mutant hearts showing abnormal great vessel alignment in mutant hearts with DORV and PTA. The aorta (AO) and pulmonary artery (PA) are indicated. (C) Haematoxylin and Eosin-stained sections of E16.5 hearts showing septation defects, including atrial septal defect (ASD) and ventricular septal defect (VSD), in mutant hearts. Scale bar: 750 µm. (D) Optical projection tomography (OPT) sections and 3D surface renderings of representative E18.5 control and mutant hearts, confirming VSD and conotruncal defects, including PTA, in mutant hearts. RA, right atrium; LA, left atrium; RV, right ventricle; LV, left ventricle.

**Table 1. Frequency of cardiac defects in *Ezh2^fl/fl;Isl1-cre* embryos at E16.5**

| Defect | Frequency of defects | |
|---|---|---|
| | *Ezh2^fl/fl* | *Ezh2^fl/fl;Isl1*-Cre |
| DORV | 0 (0%) | 2 (13%) |
| PTA | 0 (0%) | 3 (20%) |
| ASD + VSD | 0 (0%) | 3 (20%) |
| ASD + VSD + DORV | 0 (0%) | 3 (20%) |
| ASD + VSD + PTA | 0 (0%) | 1 (7%) |
| No defect | 15 (100%) | 3 (20%) |
| Total embryos | 15 | 15 |

observed in sections and 3D reconstructions from optical projection tomography of E18.5 mutant hearts (Fig. 2D), and in histological sections at postnatal day 0 (P0) (Fig. 3A,B). At P0, the right ventricular myocardium appeared less compact (Fig. 3A,B), and the aortic and mitral valve leaflets appeared thickened and irregular (Fig. S2), consistent with impaired endocardial cushion remodelling (Fig. S1B).

Ventricular morphometry further indicated that loss of Ezh2 in the Isl1 lineage altered cardiac growth in a stage-dependent manner. At E16.5, the right, but not left, ventricle diameter was slightly increased relative to controls (Fig. 3C,D). The right ventricular wall was thinner compared to controls, whereas the interventricular septum and left ventricular wall thickness were unaltered (Fig. 3C,D). In contrast, at

P0, the thickness of the right and left ventricular wall, and the IVS, was increased (Fig. 3C,D), suggesting hypertrophic remodelling. Despite robust recombination (Fig. 1), structural defects were not detected in 20% of E16.5 mutant hearts examined, suggesting incomplete penetrance, and possibly reflecting subtle phenotypes not captured in our analysis.

Outflow tract and septal defects develop similarly upon Ezh2 inactivation in Nkx2-5-labeled progenitors (Chen et al., 2012). Nkx2-5-labeled progenitors constitute a broader population within the cardiac lineage contributing cardiomyocytes of both right and left ventricles and, less prominently, endothelial cells (Ma et al., 2008). Our results indicate that loss of Ezh2 within Isl1-lineage progenitors is sufficient to produce conotruncal and septation defects.

### Ezh2 is required in Isl1-derived progenitors for normal hindlimb skeletal morphogenesis

Isl1-expressing progenitors include a subset of lateral plate mesoderm derivatives that contribute to hindlimb development (Yang et al., 2006). At E16.5, hindlimbs were smaller in *Ezh2* mutant embryos compared with controls (Fig. 4A). Whole-mount skeletal preparations at P0 stained with Alcian Blue and Alizarin Red showed hindlimb skeletal malformations (Fig. 4B-D). The pelvic girdle was abnormal, with marked ischial hypoplasia or absence (Fig. 4C). Distal hindlimb patterning was also altered, with malformation and fusion of tarsal elements, including the calcaneus and talus (Fig. 4D). In addition, posterior digits were reduced in size and showed decreased ossification

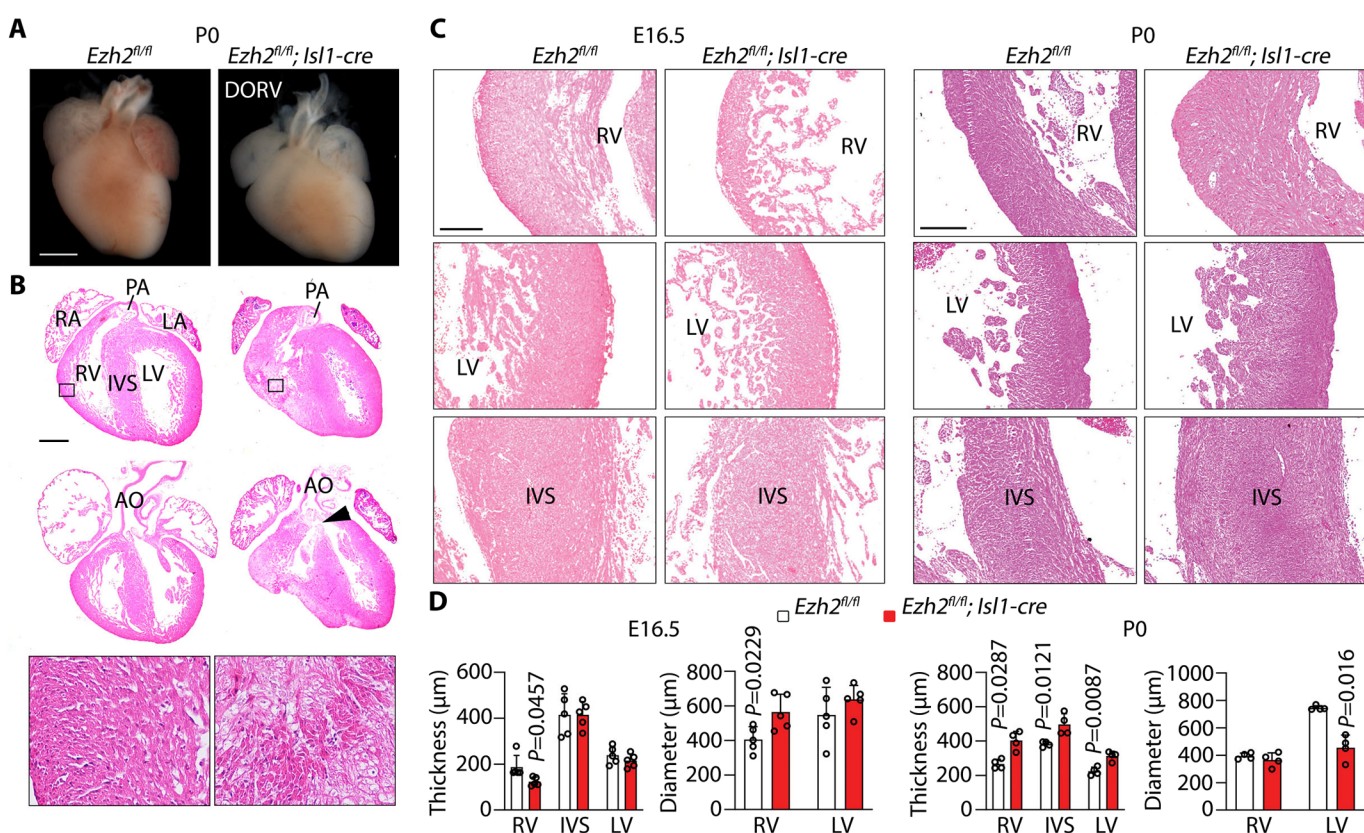

**Fig. 3. Cardiac morphology and ventricular morphometry in control and *Ezh2* mutant hearts at late gestation and birth.** (A) Gross morphology of representative P0 hearts from control (*Ezh2^fl/fl*) and mutant (*Ezh2^fl/fl;Isl1-cre*) pups, showing DORV in the mutant heart. Scale bar: 1000 μm. (B) Haematoxylin and Eosin-stained coronal sections of P0 control and mutant hearts showing the position of the PA and AO. Arrowhead points to a ventricular septal defect. Bottom micrographs are close-ups showing disorganized RV myocardium in mutants compared to controls. Scale bar: 500 μm. (C) Haematoxylin and Eosin-stained sections of RV, LV, and interventricular septum (IVS) from control and mutant hearts at E16.5 and P0. Scale bars: 150 μm. (D) Quantification of ventricular wall thickness and chamber diameter in control and mutant hearts at E16.5 and P0. Error bars denote the mean ±s.d. of five hearts per genotype, analysed by two-tailed unpaired *t*-test with Welch's correction.

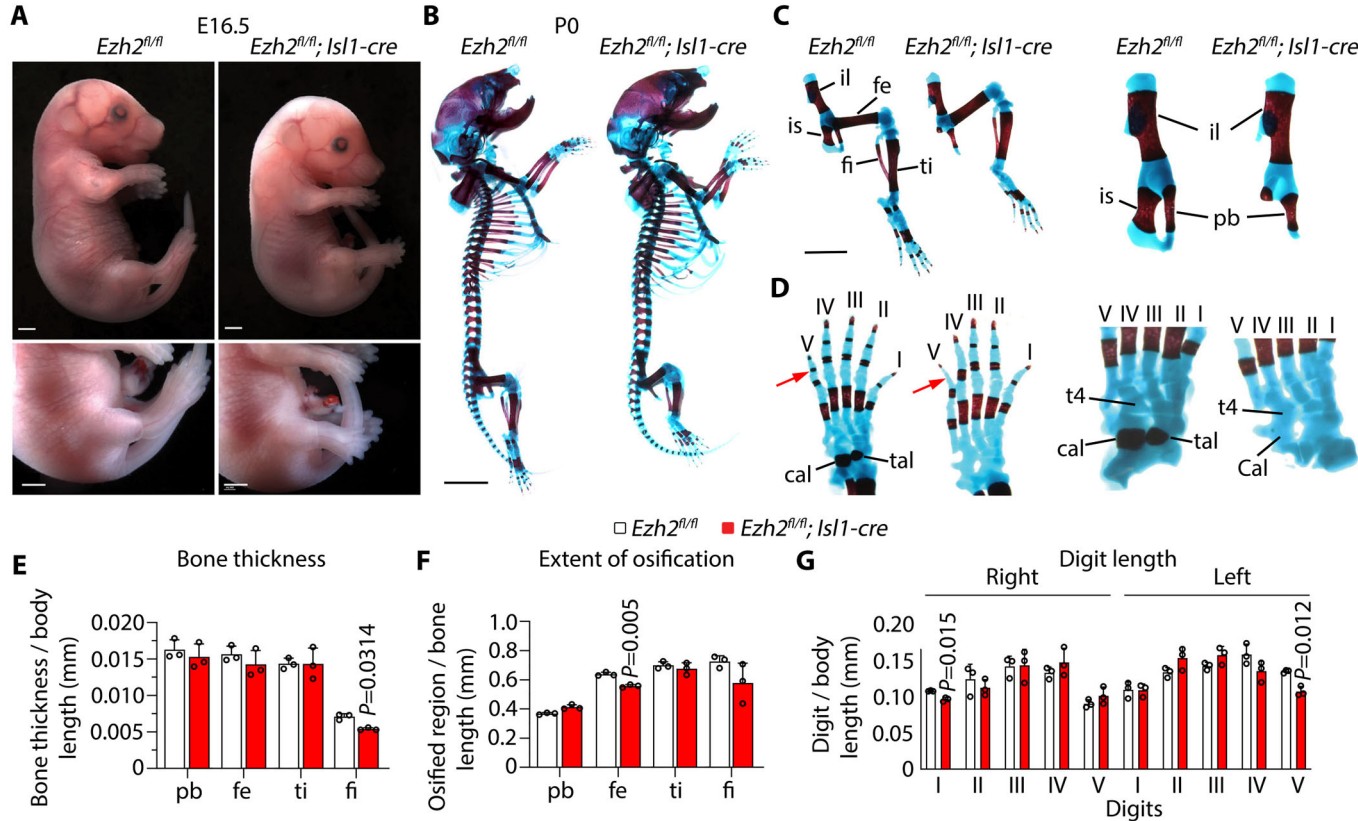

**Fig. 4. *Ezh2* inactivation in Isl1-expressing progenitors disrupts hindlimb skeletal morphogenesis.** (A) Gross morphology of representative E16.5 control (*Ezh2^fl/fl^*) and mutant (*Ezh2^fl/fl^;Isl1-cre*) embryos showing hindlimb differences. Bottom panels show higher magnification views of the hindlimb. Scale bars: 1000 µm. (B) Whole-mount skeletal preparations of P0 control and mutant pups stained with Alcian Blue and Alizarin Red. Scale bar: 2500 µm. (C) Higher magnification views of hindlimb from P0 skeletal preparations showing femur (fe), tibia (ti), fibula (fi), and ilium/pubic region. Ilium (il), ischium (is), and pubic bone (pb) are indicated. Scale bar: 2000 µm. (D) Higher magnification views of feet from skeletal preparations showing tarsal elements and digits. Arrows indicate abnormal tarsal morphology/fusion. Tarsal 4 (t4), calcaneus (cal), and talus (tal) are indicated. (E-G) Quantification of hindlimb skeletal measurements in control and mutant pups, including bone thickness normalized to body length (E), extent of ossification (F), and digit length normalized to body length (G). Error bars denote the mean±s.d. of limbs of five embryos per genotype, analysed by two-tailed unpaired *t*-test with Welch's correction.

relative to controls, consistent with impaired distal patterning and/or delayed mineralization (Fig. 4D). Quantitative analyses supported these observations and showed reduced fibular thickness and reduced femoral ossification in mutant pups (Fig. 4E,F). In contrast, forelimb skeletal development remained comparable between genotypes, with no differences detected in forelimb measurements (Fig. S3A). Isl1-expressing progenitors where β-catenin signalling regulates skeletogenesis are required for jaw development (Akiyama et al., 2014). The jaw appeared normal in *Ezh2* mutants, suggesting that Ezh2 does not have a major function in development of derivatives of Isl1-progenitors from the pharyngeal mesoderm. However, these progenitors also contribute to branchiomeric muscle development (Nathan et al., 2008). Systematic analysis of craniofacial structures in mutants is necessary to assess the requirement of Ezh2 in pharyngeal mesoderm development. Body weight and length were similar between genotypes at E16.5, whereas P0 body weight was mildly decreased in mutants (Fig. S3B), consistent with compromised perinatal development.

### Conditional inactivation of Ezh2 in Isl1-expressing progenitors results in perinatal lethality

At birth (P0), *Ezh2* mutants appeared cyanotic, consistent with severe cardiopulmonary compromise (Fig. 5A). Genotype frequencies assessed across embryonic stages (E10.5, E12.5,

E15.5, E16.5, E17.5) and P0 were consistent with expected Mendelian ratios during gestation, indicating preserved embryonic viability (Fig. 5B). In contrast, survival dropped sharply at birth, and no mutant pups survived beyond P0 (Fig. 5C,D). Together, these findings indicate that Isl1-lineage inactivation of Ezh2 caused perinatal lethality, consistent with conotruncal and septal defects.

Our results show that Ezh2 activity in Isl1-lineage progenitors is required for coordinated development of cardiac and hindlimb structures. Loss of Ezh2 in this lineage was sufficient to produce a highly penetrant spectrum of conotruncal and septation defects with abnormal ventricular remodelling at birth, accompanied by hindlimb skeletal malformations affecting the pelvis, tarsal elements, and distal ossification. The combination of severe structural CHDs, neonatal cardiac remodelling, cyanosis at birth, and complete perinatal lethality indicates that Ezh2 in Isl1-derived progenitors is essential for developmental programs that sustain cardiopulmonary function immediately after birth.

### DISCUSSION
In this study, lineage tracing confirmed Isl1-cre lineage contribution to second heart field derivatives, the neural tube, pharyngeal region, and hindlimb buds during mid-gestation. Inactivation of Ezh2 in Isl1-expressing progenitors caused CHDs and hindlimb skeletal

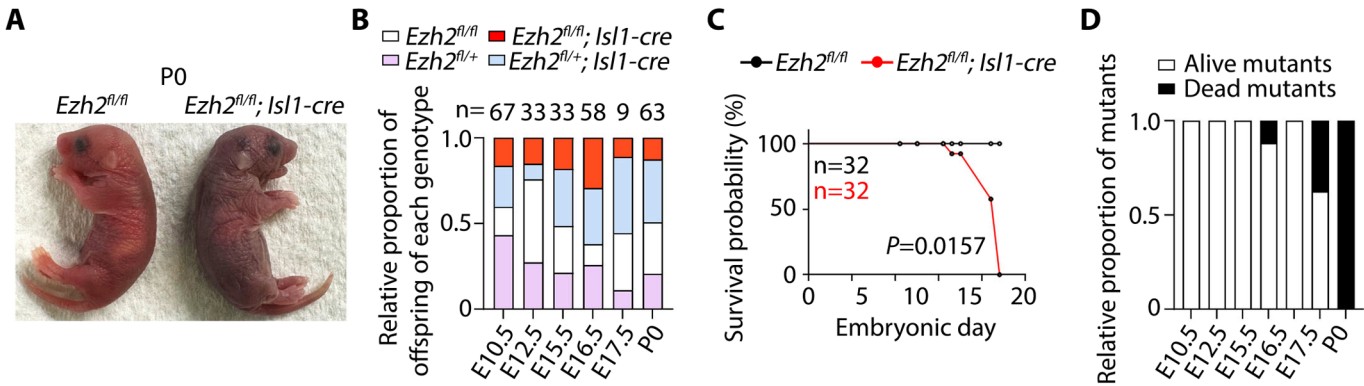

**Fig. 5. *Ezh2* inactivation in Isl1-expressing progenitors causes perinatal lethality.** (A) Representative P0 control (*Ezh2*<sup>fl/fl</sup>) and mutant (*Ezh2*<sup>fl/fl</sup>;*Isl1-cre*) pups. (B) Relative proportion of offspring genotypes at the indicated embryonic stages and at P0 from crosses of *Ezh2*<sup>fl/+</sup>;*Isl1-cre* males and *Ezh2*<sup>fl/fl</sup> females. Numbers above bars indicate total embryos or pups analysed at each stage. (C) Kaplan–Meier survival curve showing survival probability of control and mutant offspring during embryonic development and at birth. Sample size (*n*) is indicated. (D) Relative proportion of mutant offspring alive or dead at the indicated stages.

malformations associated with perinatal lethality, supporting a requirement for Ezh2 in development of Isl1-lineage derivatives.

Ezh2 inactivation using Nkx2-5-cre previously established its requirement for cardiogenesis, with loss of Ezh2 impairing cardiomyocyte proliferation and myocardial compaction and producing septation defects associated with defective endothelial-to-mesenchymal transition (Chen et al., 2012; He et al., 2012). Nkx2-5-cre is active in an early cardiac progenitor population contributing to derivatives of both the first and second heart field (George et al., 2015; Ma et al., 2008). Nkx2-5-driven Ezh2 inactivation caused defects that prominently affected outflow tract alignment and septation, together with abnormalities of myocardial growth and compaction (Chen et al., 2012; He et al., 2012). In our study, inactivation of Ezh2 in the Isl1 lineage was sufficient to drive conotruncal and septation defects, including DORV and PTA, together with ASDs and VSDs, that persisted to birth. Thickened and irregular aortic and mitral valve leaflets at P0 further support abnormal endocardial cushion remodelling.

At E16.5, right ventricular wall thickness was reduced, and right ventricular chamber diameter was increased, while left ventricular wall thickness and interventricular septal thickness remained comparable to controls. By birth, wall thickness increased in both ventricles and the interventricular septum, suggesting hypertrophic remodelling. This could reflect failure of the heart to adapt to rapidly increasing physiological demands during the transition from the end of gestation to birth. Ezh2 inactivation in anterior second heart field progenitors using Mef2c-AHF-cre, which contributes largely to right ventricular and outflow tract myocardium (Verzi et al., 2005), causes postnatal right ventricular hypertrophy (Delgado-Olguin et al., 2012), supporting a requirement for Ezh2 in regulating postnatal growth responses in second heart field derivatives. Ezh2 also stabilizes cardiac gene expression in differentiating myocardium, in part by repressing the homeodomain transcription factor *Six1* and downstream skeletal muscle gene expression (Delgado-Olguin et al., 2012; He et al., 2012), but how these activities integrate across distinct cardiac progenitor populations remains incompletely understood. The prominence of outflow tract and right ventricular abnormalities after Ezh2 inactivation using Nkx2-5-cre (Chen et al., 2012; He et al., 2012) and Isl1-cre suggest a major function of Ezh2 in development of second heart field derivatives. Molecular profiling of Ezh2-deficient Isl1-lineage derivatives during development is required to fully understand the function of Ezh2 in cardiac development and hypertrophic response.

Incomplete penetrance of overt structural defects at E16.5, despite robust Isl1-cre activity, is consistent with the variable expressivity of conotruncal and septation phenotypes (Benson et al., 1998; Dupays et al., 2019) and could also reflect subtle abnormalities not captured by our analysis. Outside the heart, Isl1-lineage inactivation of Ezh2 caused fully penetrant hindlimb abnormalities. Hindlimbs were smaller by E16.5, and skeletal preparations at birth showed pelvic malformations with ischial hypoplasia or absence, malformation and fusion of tarsal elements, reduced size of posterior digits, and decreased ossification. Forelimb development remained comparable between genotypes, consistent with restricted Isl1-lineage contribution to the hindlimb (Chi et al., 2023; Suzuki et al., 2012; Yang et al., 2006). Most prior limb studies deleted Ezh2 broadly in limb mesenchyme or in skeletal lineages, for example, using Prrx1-Cre, Col2a1-Cre, or Osx/Sp7-Cre (Deimling et al., 2018; Lui et al., 2016; Pal et al., 2021; Wyngaarden et al., 2011). In contrast, Isl1-cre restricts Ezh2 inactivation to a smaller progenitor compartment, affecting pelvic, tarsal, and posterior digits. The overlap between posterior defects in Ezh2-Isl1-cre and the anteroposterior and proximodistal patterning abnormalities reported after Prx1-cre-mediated Ezh2 inactivation is consistent with discoordinated limb patterning and subsequent ossification (Deimling et al., 2018; Wyngaarden et al., 2011).

The combination of conotruncal malformation and septation defects overlaps with phenotypes in models perturbing signalling pathways that govern second heart field morphogenesis and outflow tract alignment, including BMP, Wnt/β-catenin, and FGF pathways (Yang et al., 2006). Because Isl1-cre targets progenitor populations that seed both the anterior second heart field and the hindlimb, the coexisting conotruncal and hindlimb phenotypes are consistent with Ezh2 modulating shared pathways in Isl1-derived progenitors linked to heart and limb malformations (Yang et al., 2006). In early limb bud mesenchyme, disrupting Ezh2-mediated anteroposterior patterning of *Hox* expression domains leads to shortened proximodistal limb segments (Wyngaarden et al., 2011). In osteogenic differentiation assays using mesenchymal stromal cells, reducing Ezh2 activity promoted osteogenic differentiation accompanied by upregulation of *MSX2* and *HEY1*, which are downstream targets of BMP and Wnt signalling (Dudakovic et al., 2015). Consistent with a requirement for Ezh2 in limb mesenchyme coordinating musculoskeletal development, limb mesenchyme–restricted Ezh2 inactivation altered skeletal patterning (Pal et al., 2021). Together with the overlap between cardiac phenotypes found in this study and those produced by

perturbation of BMP, Wnt/β-catenin, and FGF pathway components, our findings suggest that Ezh2 deficiency in Isl1-lineage derivatives could alter transcriptional responsiveness to BMP and Wnt/β-catenin.

Perinatal lethality coincided with cyanosis, while Mendelian genotype distributions during gestation indicated preserved embryonic viability, supporting requirement for Isl1-lineage Ezh2 function at the transition to postnatal life. Conotruncal and septation defects could contribute to neonatal collapse, but these were not fully penetrant in *Ezh2* mutants. While subtle cardiac defects could have gone undetected in our analysis, contribution of additional Isl1-lineage derivatives remains possible. Isl1-expressing progenitors contribute to the proepicardium, with Isl1-lineage cells expressing proepicardial markers such as *Wt1* and *Tbx18* (Zhou et al., 2008). Accordingly, Isl1-cre activity was detected in epicardial cells (Fig. 1B). WT1-cre-driven Ezh2 inactivation leads to embryonic lethality with myocardial hypoplasia and a reduced coronary vascular plexus (Jiang et al., 2023). Therefore, impaired epicardium development could contribute to late-gestation right ventricular thinning and decompensation of *Ezh2* mutant neonates. Extracardiac contributions are also possible. For example, *Ezh2* inactivation in developing lung epithelium decreases airway branching and volume, leading to demise at birth (Galvis et al., 2015). Gene expression profiling of Isl1-lineage derivatives during embryogenesis and at birth will be required to uncover underlying mechanisms of Ezh2 action. Coupled phenotyping of coronary vasculature and lung maturation would help distinguish primary transcriptional dysregulation in specific Isl1-derived compartments from secondary consequences of structural malformations leading to perinatal lethality.

Our findings establish the requirement for Ezh2 within Isl1-lineage progenitors for cardiac and hindlimb development and for survival at birth.

## MATERIALS AND METHODS
### Mice
All animal experiments were approved by the Animal Care Committee at the Toronto Centre for Phenogenomics. The following mouse strains were used: *Ezh2fl/fl* (*129P2/OlaHsd-Ezh2tm1Tara*) (O'Carroll et al., 2001), *Isl1-cre* (*Stock-Isl1tm1(cre)Cos*) (Yang et al., 2006), and *ROSA26mTmG* (*Stock-Gt(ROSA)26Sortm4(ACTB-tdTomato,-EGFP)Luo/J*) (Muzumdar et al., 2007). All mouse strains were maintained in a C57BL6/J background. Mice were housed in standard vented cages supplied with 100% fresh air under 12 h light-dark cycles. Room temperature was maintained between 20-22°C, and humidity between 40-60%. Food and water were provided *ad libitum*; mice were fed standard chow (Teklad Global 18% Protein Rodent Diet, ENVIGO, 491 TD.2918X).

Female mice homozygous for the *Ezh2* allele (*Ezh2fl/fl*) were crossed with *Isl1-cre* positive males, which express cre recombinase in *Isl1*-expressing progenitors and their derivatives, to generate heterozygous *Ezh2fl/+;Isl1*-cre offspring. In the second generation, *Ezh2fl/fl* females were bred with *Ezh2fl/+; Isl1*-cre males in timed-matings to generate *Ezh2fl/+* (wild-type), *Ezh2fl/fl* (wild-type), *Ezh2fl/+;Isl1*-cre (heterozygous wild-type), and *Ezh2fl/fl;Isl1*-cre (homozygous mutant) embryos and pups. Lineage tracing experiments were conducted on mouse embryos and pups generated from breeding *Ezh2fl/+; Isl1*-cre males with *Ezh2fl/fl;ROSA26mTmG* females. Both male and female embryos were collected from *Ezh2fl/fl;ROSA26mTmG* or *Ezh2fl/fl* dams at E10.5, E12.5, and E16.5. Pups at P0 were also obtained for analysis. Mice were mated in the evening, and the presence of a vaginal plug was assessed the following morning. The day of plug detection was designated as embryonic day (E) 0.5. All mice were genotyped from ear notches or yolk sacs as described (Vuong and Delgado-Olguin, 2018) using the primers in Table S1.

### Haematoxylin and Eosin staining
Mouse fetal hearts were collected at E16.5, E18.5, and pups at P0 for histological analysis. Hearts were dissected in ice-cold 1X PBS and fixed in 4% paraformaldehyde overnight at 4°C. The next day, hearts were washed with 1X PBS for 10 min three times at room temperature and stored in 70% ethanol overnight at 4°C. Hearts were dehydrated in an ethanol series: twice each in 70%, 85%, and 95% ethanol for 30 min. Samples were then cleared using 50% ethanol: xylene for 5 min and 100% xylene 3× for 10 min. Tissues were left at 60°C in 50% xylene: wax for 30 min and incubated at 37°C overnight. The next day, tissues were placed at 60°C until the wax melted. Afterwards, the wax was changed every 1 h at 60°C for a total of four times. Tissues were then embedded in paraffin for sectioning. 5 µm cross-sections were obtained, mounted on glass slides, and stored at room temperature. Sections were de-waxed at 60°C overnight and dehydrated in another ethanol series before Haematoxylin and Eosin staining. Sections were cleared in xylene and imaged using a Nikon Eclipse Ni-U microscope. Images were captured using NIS Elements software (Nikon Instruments Inc., Melville, New York, USA).

### Microscopy
To examine the whole heart morphology, hearts were dissected in ice-cold 1X PBS. Extracardiac membranes were removed to reveal a clear view of the outflow tract. Hearts were imaged on petri dishes coated with black Sylgard using a Nikon SMZ1000 Stereomicroscope and NIS elements software (Nikon Instruments Inc., Melville, New York, USA).

### Whole-mount skeletal staining
To examine the skeletal defects in *Ezh2fl/fl;Isl1-cre* embryos, whole-mount skeletons were stained as previously reported (Rigueur and Lyons, 2014) with minor modifications. Mouse embryos were collected and placed in 1X PBS on ice. Viscera and skin were removed. Specimens were fixed overnight in 70% ethanol at 4°C, then dehydrated in 95% ethanol for 1 h, followed by acetone overnight at room temperature. Following dehydration, embryos were placed in 0.03% Alcian Blue solution (30 mg Alcian Blue in 80% ethanol and 20% glacial acetic acid) until cartilage was completely stained (~6-7 h). Alcian Blue was replaced with 0.005% Alizarin Red solution (5 mg Alizarin Red in 1% KOH) overnight at room temperature. Embryos were cleared in 1% KOH for 7 h and transferred to 50% glcyerol:50% (1%) KOH until tissue appeared transparent. Specimens were then transferred to 100% glycerol for long-term storage.

To examine skeletal defects in P0 pups, after euthanasia, pups were first placed in 70°C ddH₂0 for 30 s to facilitate skin removal. Specimens were then fixed overnight in 70% ethanol at 4°C, dehydrated in 95% ethanol for 1 h and acetone overnight at room temperature. Cartilage was stained with 0.03% Alcian Blue overnight at room temperature. Excess staining from tissues was removed by washing with 70% ethanol for 15 min twice at room temperature and incubating in 95% ethanol overnight. Tissues were pre-cleared using 1% KOH for 1 h at room temperature, and ossified tissue was stained with 0.005% Alizarin Red solution for 4 h or until completely stained at room temperature. Specimens were cleared using 50% glycerol: 50% (1%) KOH until transparent and moved to100% glycerol for long-term storage. Stained skeletons were imaged using a Nikon SMZ1000 Stereomicroscope.

### Immunofluorescence
Mouse embryos were dissected in 1X PBS and fixed in 4% PFA for 2 h at 4°C. Following fixation, embryos were washed three times with 1X PBS for 10 min at room temperature and placed in 15% sucrose dissolved in DEPC-PBS overnight at 4°C on a rotator. The solution was replaced with 30% sucrose overnight. Embryos were embedded in O.C.T. Compound (Tissue-Tek), and 10 µm frozen sections were mounted on glass slides. Sections were fixed with 4% PFA for 10 min at room temperature, and slides were washed with 1X PBS three times, 5 min each. Sections were blocked with 5% goat serum, 3% bovine serum albumin, and 0.27% Tween-20 in 1X PBS for 2 h at room temperature. Sections were stained with Anti-GFP (Abcam, ab13970, 1:1000) overnight at 4°C and visualized using the Alexa Fluor™ 488 conjugated goat anti-chicken (Invitrogen, A-11039, 1:700) secondary antibody.

### Western blot
E16.5 mouse embryos were dissected in cold 1X PBS-DEPC. Right and left ventricles were pooled for each replicate, and tissues were homogenized in

500 µl of Tissue Extraction Reagent (Thermo Fisher Scientific, FNN0071) containing 50 µl/ml Protease Inhibitor Cocktail (Millipore Sigma, P2714) using a bead homogenizer (TissueLyser II) at 30kHz for 1 min. Samples were then sonicated for 1 min and centrifuged at 13,000 $g$ for 5 min. Supernatant was collected, and protein concentration was measured using the Pierce BCA Protein Assay Kit (Thermo Fisher Scientific, 23227). Lysates were resolved by SDS–PAGE and transferred to a PVDF membrane using the Trans-Blot Turbo Transfer System (Bio-Rad). Membranes were blocked in 3% BSA in TBST and then incubated overnight at 4°C with primary antibodies in TBST containing 1% BSA. Membranes were then washed three times and incubated with secondary antibodies in TBST containing 1% BSA for 1h. Images and the relative intensities of the immunoreactive bands were captured using the Odyssey Fc system (LI-COR Biotechnology). Primary antibodies used were anti-H3K27me3 (Active Motif, 39155, 1:800), H3 (Abcam, a61791, 1:800), Ezh2 (Cell Signalling, 5246, 1:800), Vinculin (Cell Signalling, 13901, 1:700), H2aK119ub (Cell Signalling, 8240, 1:800), and H2A (Cell Signalling, 12349, 1:800). The following secondary antibodies were used: mouse anti-rabbit IgG–HRP (Santa Cruz Biotechnology, sc-2357, 1:1000) and m-IgG Fc BP–HRP (Santa Cruz Biotechnology, sc-525409, 1:1000). 30 ug of protein were loaded per replicate for each experiment. Protein levels of H3K27me3 were normalized to H3, H2aK119ub to H2A, and Ezh2 to Vinculin, respectively. All specimens analysed in each experiment were run in the same gel.

## Optical projection tomography (OPT)

OPT was performed as reported (Sharpe et al., 2002). Embryonic hearts were dissected in cold PBS and fixed in 4% PFA at 4°C. After fixation, hearts were washed in PBS, embedded in low-melting-point agarose, and mounted on the OPT sample holder. Agarose blocks containing individual hearts were dehydrated through a graded methanol series and optically cleared in a benzyl alcohol: benzyl benzoate (BABB). Projection datasets were reconstructed with Amira for 3D visualization.

## Acknowledgements
We thank The Centre for Phenogenomics for mouse husbandry, research support, and dedicated animal care by veterinary professionals; and Paul Paroutis and Kimberly Lau at the SickKids Imaging Facility for microscopy support.

## Competing interests
The authors declare no competing or financial interests.

## Author contributions
Conceptualization: P.D.-O.; Data curation: H.A., D.B.P., L.C., A.N.C., P.D.-O.; Formal analysis: H.A., D.B.P., L.C., A.N.C.; Investigation: H.A., D.B.P., L.C., A.N.C.; Methodology: H.A., D.B.P., L.C., A.N.C.; Project administration: P.D.-O.; Supervision: A.N.C., P.D.-O.; Validation: H.A., L.C.; Visualization: H.A.; Writing – original draft: H.A., P.D.-O.; Writing – review & editing: H.A., P.D.-O.

## Funding
P.D.-O. is funded by the Canadian Institutes of Health Research (CIHR) (grant nos. 162208, 149046, and 468633), and the Heart & Stroke Foundation of Canada (grant no. G-17-0018613). Open Access funding provided by University of Toronto. Deposited in PMC for immediate release.

## Data and resource availability
All relevant data and details of resources can be found within the article and its supplementary information.

## First Person
This article has an associated First Person interview with the first author of the paper.

## Peer review history
The peer review history is available online at https://journals.biologists.com/bio/lookup/doi/10.1242/bio.062550.reviewer-comments.pdf

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
