## [Peer Review File · Biology Open]

Enhancer of zeste homolog 2 (Ezh2) is required in mouse Isl1-expressing progenitors for proper development of cardiac and skeletal hindlimb structures

Hamna Ammar, Daniel B. Patolsky, Lijun Chi, Amalia N. Caballero and Paul Delgado Olguin

DOI: 10.1242/bio.062550

Editor: Tristan A Rodriguez

Review timeline

Original submission:	27 February 2026
Editorial decision:	9 March 2026
First revision received:	14 March 2026
Accepted:	16 March 2026

Original submission

First decision letter

MS ID#: bio.062550

MS Title: Enhancer of zeste homolog 2 (Ezh2) is required in mouse Isl1-expressing progenitors for proper development of cardiac and skeletal hindlimb structures

Authors: Hamna Ammar, Daniel B. Patolsky, Lijun Chi, Amalia N. Caballero and Paul Delgado Olguin

I have now reached a decision on the above manuscript.

The reviewer reports are shown at the bottom of this email.

As you will see, the reviewers raised a number of substantial criticisms that prevent me from accepting the paper at this stage.

They suggest, however, that a revised version will likely prove acceptable, if you can address their concerns. If you think that you can deal satisfactorily with the criticisms on revision, I would be pleased to see a revised manuscript. There is no requirement for additional experimentation. However, should additional experimental results be available that address the mechanistic concerns of the reviewers then please provide these in the revised manuscript.

At this stage, we also ask you to ensure your manuscript complies with our formatting guidelines. Provided you are able to fully address the referees' comments, we are positive about publication of your paper (we accept over 95% of revision submissions) and therefore hope you won't mind any extra work involved in reformatting your manuscript at this point.

Please upload both a 'clean' version of your Word file, along with a highlighted version clearly showing where you have made changes in the revised manuscript. Please avoid using 'Track changes' in Word files as these are lost in PDF conversion.

I should be grateful if you would also provide a point-by-point response detailing how you have dealt with the points raised by the reviewers in the 'Response to Reviewers' box. Please attend to

all of the reviewers' comments. If you do not agree with any of their criticisms or suggestions please explain clearly why this is so.

Reviewer 1

Comments for the author

In this manuscript, Amar et al describe the developmental phenotype in the mouse caused by the conditional loss of *Ezh2*, when driven by an *Isl1*-Cre driver. The work is carefully conducted, and the results shown are of high quality and robustness to properly support their conclusions.

EZH2 drives the function of Polycomb complex 2 (PRC2), and its role in development has been previously studied in various context, some of them by the senior authors of this work. PRC2 is in charge of establishing facultative heterochromatin by tri-methylation of lysine 27 from Histone 3 (H3K27me3), and largely endowed with the repression of alternate lineage specific genes to assure robust cell identity. Therefore, it is hardly surprising that its deletion in any cell type, lineage or organ system leads to obvious developmental defects.

Here, the authors use a well characterized *Isl1*-Cre mouse line, that drives recombination in second heart field progenitors and in the lateral plate mesoderm, from which the hindlimb will form. Therefore, it is hardly surprising that they find defects in these lineages. While true that "...EZH2 as a regulator of ISL1-lineage progenitor development" (line 29), maybe this reads a bit too specific. In this regard, the driver is also highly active in neural crest derivatives from the first and second branchial arch, but no specific defects related to this are described. Is this because the authors did not examine it, or were they not fund? At least a mention should be made.

Also, at some point the authors seem to over-interpret their findings, as the mention of "...coordinated cardiac and hindlimb morphogenesis" in the Abstract, Introduction and Results, while might be true, is not something that can be concluded from the experiments and data shown here. The presence of defects in heart and limb only reflects that the Cre driver is expressed in these tissues.

If something is felt missing is some more discussion on the possible mechanisms underlying the phenotypes observed here. As EZH2 mediates PRC2 gene repression, one would surmise that improper gene expression is responsible for the defects. Have the authors explored in any way this issue?

Related to this point, and although not required for this manuscript, it would be very interesting to study the transcriptional changes caused by *Ezh2* loss in *Isl1*+ cardiac progenitors, and how these changes compare to those caused by the deletion of *Ezh2* using an *Nkx-2.5* Cre driver (Delgado-Olguin et al., 2012).

Reviewer 2

Comments for the author

The manuscript by Ammar et al. analyzed mutant mice, with a conditional ablation of the Enhancer of zeste homolog 2 (*Ezh2*) using an *Isl1*-Cre driver. The authors observed a series of outflow tract and atrial and ventricular septal defects, but there were also about 20% of mutants which did not develop a structural heart defect. Analysis of the mutant perinatally also observed some alterations in the ventricular wall with some increase in wall thickness. No mutant animal survived birth, however, the reason for this fully penetrant phenotype remains unexplained and is unlikely to be based on the cardiac phenotype, given that animals with a normal heart were found at an earlier time points. In addition to the heart there were also alterations in the patterning and ossification of the hindlimb.

Overall, this is a nice descriptive paper analyzing the embryological consequences of *Ezh2* ablation in mice. The data are solid and well presented. However, it is unfortunately not well explained why

the heart phenotype is not fully penetrant, yet all mutant animals die at birth. The study fails to identify the cause of this lethality.

Previous studies ablating *Ezh2* using *Nkx2.5* observed aberrant expression of *Six1* a transcription factor which drives skeletal muscle. Given that alterations in epigenetic silencing are likely also present in the current *Ezh2* mutant, it is surprising that the authors didn't check what is known from other studies. Thus, please check in your mutant whether *Six1* is misexpressed and whether aberrant skeletal muscle gene expression is present.

Isl1 is known to be expressed in the proepicardium and subsequently also in the epicardium. You also see an expression in the epicardium (Fig. 1B) but fail to mention this. Could you please also please discuss whether the hypoplastic right ventricular phenotype at E16.5 might relate to a failure in the epicardium. In this regard, do you know if the coronary vasculature is normally developed?

Minor

Fig.1 Please revise the heading. Two-thirds of the figure are about *Isl-Cre* lineage tracing and not about the *Isl-1*-mediated conditional KO of *Ezh2*.

Fig. 3D: diameter - graph legend y-axis.

Reviewer's Responses to Questions

Experimental quality

Does each figure have the proper controls?

If 'No', please indicate reasons in Comments for Author box below.

Reviewer #1:

- Yes

Reviewer #2:

- Yes

Were the data analyzed using appropriate statistical tests?

If 'No', please indicate reasons in Comments for Author box below.

Reviewer #1:

- Yes

Reviewer #2:

- Yes

Reproducibility

Were experiments performed using adequate number of biological replicates?

If 'No', please indicate reasons in Comments for Author box below.

Reviewer #1:

- Yes

Reviewer #2:

- Yes

Does the methods section provide sufficient detail to permit reproducibility?

If 'No', please indicate reasons in Comments for Author box below.

Reviewer #1:

- Yes

Reviewer #2:

- Yes

Completeness

Are the manuscript's conclusions supported by the data?

If 'No', please indicate reasons in Comments for Author box below.

Reviewer #1:

- Yes

Reviewer #2:

- No

Scholarship

Do the authors cite and discuss the merits of data that would argue for and against their conclusion?

If 'No', please indicate reasons in Comments for Author box below.

Reviewer #1:

- Yes

Reviewer #2:

- No

Does the manuscript title & abstract accurately reflect the contents of the manuscript, without hyperbole?

If 'No', please indicate reasons in Comments for Author box below.

Reviewer #1:

- Yes

Reviewer #2:

- Yes

First revision**Author response to reviewers' comments**

We thank the reviewers for the thorough review of our manuscript and the insightful comments.

Reviewer 1

Here, the authors use a well characterized *Isl1*-Cre mouse line, that drives recombination in second heart field progenitors and in the lateral plate mesoderm, from which the hindlimb will form. Therefore, it is hardly surprising that they find defects in these lineages. While true that "...*EZH2* as a regulator of *ISL1*-lineage progenitor development" (line 29), maybe this reads a bit too specific. In this regard, the driver is also highly active in neural crest derivatives from the first and second branchial arch, but no specific defects related to this are described. Is this because the authors did not examine it, or were they not found? At least a mention should be made.

Also, at some point the authors seem to over-interpret their findings, as the mention of "...coordinated cardiac and hindlimb morphogenesis" in the Abstract, Introduction and Results, while might be true, is not something that can be concluded from the experiments and data shown here. The presence of defects in heart and limb only reflects that the Cre driver is expressed in these tissues.

Agreed, potential functions of *Ezh2* in additional *Isl1*-cre-expressing progenitors, including neural crest derivatives from the first and second branchial arch, cannot be ruled out from our

phenotyping focused on heart and hind limb. Given the grossly normal appearance of the jaw in mutant embryos, we did not systematically analyze craniofacial development. We have now corrected the omission of this important information (lines 154-160) and indicate that investigating a potential function of *Ezh2* in derivatives of pharyngeal mesoderm would require analysis of craniofacial structures.

Accordingly, we also revised the statement regarding the requirement of *Ezh2* for coordinated cardiac and hindlimb morphogenesis in lines 70-72 and 269-270 to reflect the consistency between the activity of the cre driver and affected structures. In the abstract (lines 28-29), we toned down the finishing statement. It now reads: "Together, these findings identify a requirement for *EZH2* in *Isl1*-lineage contribution to heart and hindlimb development."

If something is felt missing is some more discussion on the possible mechanisms underlying the phenotypes observed here. As *EZH2* mediates PRC2 gene repression, one would surmise that improper gene expression is responsible for the defects. Have the authors explored in any way this issue?

Related to this point, and although not required for this manuscript, it would be very interesting to study the transcriptional changes caused by *Ezh2* loss in *Isl1*+ cardiac progenitors, and how these changes compare to those caused by the deletion of *Ezh2* using an *Nkx-2.5* Cre driver (Delgado-Olguin et al., 2012).

Yes, this is a very important point. Defining transcriptional consequences of *Ezh2* loss in *Isl1*-lineage derivatives is required to uncover underlying mechanisms. We couldn't pursue our original plan to profile gene expression and chromatin accessibility in single cells in mutant embryos. Therefore, we focused on developmental phenotyping. Because of this, we limited our discussion to established gene expression and pathways controlled by *Ezh2* that are relevant for heart development, hypertrophy, limb morphogenesis, and osteogenesis, including *Six1* and *Hox* genes, and Wnt and Bmp signaling (lines 234-250).

We also expanded the Discussion to address perinatal lethality despite incomplete penetrance of overt cardiac defects by considering subtle functional abnormalities and contribution of additional *Isl1*-lineage derivatives (lines 251-268). In this new paragraph, we note *Isl1*-lineage contribution to proepicardium/epicardium (Zhou et al., 2008) and *Isl1*-cre activity in epicardial cells (Fig. 1B), cite evidence that epicardial *Ezh2* inactivation using *Wt1*-cre impairs myocardial growth and coronary vascular development (Jiang et al., 2023), and highlight potential extracardiac contributions such as impaired lung maturation upon epithelial *Ezh2* loss (Galvis et al., 2015). We conclude by stating that unbiased gene expression profiling of *Isl1*-lineage derivatives across late gestation and birth, coupled with coronary and lung phenotyping, will be required to distinguish primary transcriptional dysregulation from secondary effects.

Reviewer 2

Overall, this is a nice descriptive paper analyzing the embryological consequences of *Ezh2* ablation in mice. The data are solid and well presented. However, it is unfortunately not well explained why the heart phenotype is not fully penetrant, yet all mutant animals die at birth. The study fails to identify the cause of this lethality.

Agreed. Cardiac malformations were not fully penetrant at E16.5, yet perinatal lethality was fully penetrant. While conotruncal and septation defects could contribute directly to neonatal collapse, subtle cardiac defects could have gone undetected. In addition, contribution of *Isl1*-lineage derivatives outside the heart remains possible, including epicardial populations that support coronary vascular development and myocardial maturation, as well as potential extracardiac defects that could exacerbate postnatal decompensation. We have revised the Discussion to address these possibilities and to state that identifying the primary driver of lethality will require more extensive developmental phenotyping together with unbiased gene expression profiling of *Isl1*-lineage derivatives across late gestation and birth (lines 251-268).

Previous studies ablating *Ezh2* using *Nkx2.5* observed aberrant expression of *Six1* a transcription factor which drives skeletal muscle. Given that alterations in epigenetic silencing are likely also

present in the current Ezh2 mutant, it is surprising that the authors didn't check what is known from other studies. Thus, please check in your mutant whether Six1 is misexpressed and whether aberrant skeletal muscle gene expression is present.

We agree with the reviewer. We couldn't pursue our original plan to profile gene expression and chromatin accessibility in single cells in mutant embryos. Therefore, we focused on developmental phenotyping. However, we have revised our manuscript to discuss established Ezh2 targets to provide plausible links to the phenotype and perinatal lethality. Moreover, we recognize the requirement of phenotyping additional structures derived from Isl1-cre lineages and gene expression profiling to uncover underlying mechanisms.

Isl1 is known to be expressed in the proepicardium and subsequently also in the epicardium. You also see an expression in the epicardium (Fig. 1B) but fail to mention this. Could you please also please discuss whether the hypoplastic right ventricular phenotype at E16.5 might relate to a failure in the epicardium. In this regard, do you know if the coronary vasculature is normally developed?

This is a very important point. We have revised the Fig. 1 and the description in the text to indicate EGFP activity in epicardial cells. We also expanded the Discussion to address perinatal lethality despite incomplete penetrance of overt cardiac defects by considering subtle functional abnormalities and contribution of additional Isl1-lineage derivatives (lines 251-268). In this new paragraph, we note Isl1-lineage contribution to proepicardium/epicardium (Zhou et al., 2008) and Isl1-cre activity in epicardial cells (Fig. 1B), cite evidence that epicardial Ezh2 inactivation using Wt1-cre impairs myocardial growth and coronary vascular development (Jiang et al., 2023), and highlight potential extracardiac contributions such as impaired lung maturation upon epithelial Ezh2 loss (Galvis et al., 2015). We conclude by stating that unbiased gene expression profiling of Isl1-lineage derivatives across late gestation and birth, coupled with coronary and lung phenotyping, will be required to distinguish primary transcriptional dysregulation from secondary effects.

Minor

Fig.1 Please revise the heading. Two-thirds of the figure are about Isl-Cre lineage tracing and not about the Isl-1-mediated conditional KO of Ezh2.

We have revised the Fig. 1 heading. It now reads:

Fig. 3D: diameter - graph legend y-axis.

Thanks for noting this. We have fixed the typo.

Second decision letter

MS ID#: bio.062550R1

MS Title: Enhancer of zeste homolog 2 (Ezh2) is required in mouse Isl1-expressing progenitors for proper development of cardiac and skeletal hindlimb structures

Authors: Hamna Ammar, Daniel B. Patolsky, Lijun Chi, Amalia N. Caballero and Paul Delgado Olguin

I am happy to tell you that your manuscript has been accepted for publication in Biology Open, pending our standard publication integrity checks. It was accepted on 16th March 2026.